# Electrical Fields in the Processing of Protein-Based Foods

**DOI:** 10.3390/foods13040577

**Published:** 2024-02-14

**Authors:** Ricardo N. Pereira, Rui Rodrigues, Zita Avelar, Ana Catarina Leite, Rita Leal, Ricardo S. Pereira, António Vicente

**Affiliations:** 1CEB—Centre of Biological Engineering, University of Minho, 4710-057 Braga, Portugal; rpereira@deb.uminho.pt (R.N.P.); ruirodrigues@ceb.uminho.pt (R.R.); zita.avelar@ceb.uminho.pt (Z.A.); ana.catarina.leite@ceb.uminho.pt (A.C.L.); rita.leal@ceb.uminho.pt (R.L.); pg42880@alunos.uminho.pt (R.S.P.); 2LABBELS—Associate Laboratory, 4710-057 Braga/Guimarães, Portugal

**Keywords:** moderate electric fields, ohmic heating, pulsed electric fields, electroporation, plant-based proteins, secondary structure, allergenicity

## Abstract

Electric field-based technologies offer interesting perspectives which include controlled heat dissipation (via the ohmic heating effect) and the influence of electrical variables (e.g., electroporation). These factors collectively provide an opportunity to modify the functional and technological properties of numerous food proteins, including ones from emergent plant- and microbial-based sources. Currently, numerous scientific studies are underway, contributing to the emerging body of knowledge about the effects on protein properties. In this review, “Electric Field Processing” acknowledges the broader range of technologies that fall under the umbrella of using the direct passage of electrical current in food material, giving particular focus to the ones that are industrially implemented. The structural and biological effects of electric field processing (thermal and non-thermal) on protein fractions from various sources will be addressed. For a more comprehensive contextualization of the significance of these effects, both conventional and alternative protein sources, along with their respective ingredients, will be introduced initially.

## 1. Introduction

Throughout recent years, there has been a noticeable aggravation of climate conditions linked to an increase in human population. In 2022, it was estimated that in 2058 the world human population will reach 10 billion [1]. The rapid growth in human population has inevitably led to an increase in productivity through agricultural expansion, a process that has further worsened the environmental crisis through increased water use, as well as resulting in greater emissions of greenhouse gases [2]. It is necessary to explore sustainable raw materials to potentially uncover sustainable protein sources, in addition to developing novel food processing technologies to achieve a socioeconomic and environmental balance, with the added benefit of improving the health of human population [3,4,5].

Protein-based foods stand as a foundation in global diets, contributing significantly to human health, nutrition, and overall well-being. The essential role of proteins as building blocks for tissues, enzymes, hormones, and various bodily functions underlines their vital significance in sustaining life. Proteins are critical macronutrients, and beyond their physiological importance, they also contribute to satiety, aiding in weight management and reducing the risk of chronic diseases. However, meeting the escalating global demand for protein-rich foods presents significant challenges, including environmental sustainability, resource scarcity, and the need for innovative production methods. The exploration of alternative protein sources, such as plant-based proteins (derived from legumes, grains, and vegetables), microorganisms (yeast, bacteria), algae, and insects, has gained traction. These sources not only offer sustainable options but also diversify dietary choices and contribute to reducing the environmental footprint of food production. As dietary preferences evolve and nutritional awareness increases, there is a growing demand for diverse, sustainable, and high-quality protein sources. New protein sources present several functional and technological challenges in food processing to ensure their successful integration into the food supply chain. Extracting or purifying proteins from non-traditional sources often requires specialized techniques that may be energy-intensive or involve complex purification processes. Adapting new textures and tastes to mimic familiar products while maintaining nutritional value is also a challenge. It is also critical to take into consideration that proteins may contain epitopes involved in allergic reactions that need to be identified and mitigated during processing to ensure product safety [6]. Achieving consistent quality and cost-effectiveness while meeting market demands is a significant hurdle, together with convincing consumers of the nutritional value and taste.

Alternative food processing technologies have emerged offering solutions to address these challenges. Traditional methods for thermal unit operations of food (such as pasteurization and sterilization) do have certain limitations. While they are effective in killing harmful bacteria and extending the shelf life of food, they can also lead to a loss of nutritional quality. This is because heat can destroy certain vitamins and nutrients. Additionally, these methods can be energy intensive, which raises concerns about their sustainability and efficiency. Newer methods of food processing are being researched to address these issues. This exploration of novel technologies promises not only to address current challenges but also to shape the future of food, revolutionizing how protein-based foods are produced, consumed, and integrated into global diets. Particular focus has been paid to physical, non-thermal techniques, such as cold plasma, high-pressure, ultrasound, and electric field technologies and irradiation treatments. The main objective of this review is to give a comprehensive overview about the electric field processing of protein-based foods covering both fundamental and applied aspects of protein science.

### 1.1. Electric Field Processing: Historical Perspective

Electric field technologies in food processing involve the application of electrical energy to modify, treat, or process food materials. This application relies on the conduction of electric currents through food material with semi-conductive properties, which is in direct contact with the electrodes, and can induce changes in food characteristics by preserving quality and enhancing functionality and safety, including extending shelf life [7]. Electric field-based technologies can be categorized into different subgroups based on their distinct action mechanisms and desired outcomes. In the realm of food processing, the techniques that exhibit greater prominence are Ohmic Heating (OH) and Pulsed Electric Field (PEF) technologies, but other variants are now emerging.

The concept of OH has been known and utilized for a considerable time, together with the development and application of pulsed electric field technology. In the early 19th century, practical uses of electrical heating surfaced through several patented innovations that capitalized on the heat-producing properties inherent in substances capable of flowing. In 1919, Anderson and Finklestein published research about milk pasteurization in a private dairy company using OH technology, recognized as the “Electro-pure process for treating milk”. During the mid-twentieth century, the introduction of PEF processing for foods took place and gained momentum following the discovery of the electroporation phenomenon in the 1950s–1960s; during the early 1960s, Heinz Doevenspeck utilized PEF to break down cells extracting fishmeal and fish oil and subsequently separated solids and liquids using a screw press [8]. PEF is considered to a be sister technology to OH; however, it is important to emphasize that the term “Ohmic heating” specifically refers to the phenomenon where heat is generated because of the resistance encountered by electric currents passing through a material. It is indeed an effect of applying an electric field, particularly when the electrical resistance within a food material causes heat generation. For example, PEF processing has the capacity to generate ohmic heating effects under specific conditions, particularly at elevated electric field strengths and extended treatment durations, especially when coupled with the high electrical conductivity of the food material.

For controlled electroheating purposes, electric fields are commonly termed as moderate electric fields (MEF) to distinguish them from PEF technology. Non-thermal effects on biomolecules, cell structure, microorganisms, and enzymes have been also observed with MEF protocols. This discovery has paved the way for an array of enhanced food processing strategies towards the extraction and functionalization of biomolecules. Presently, OH and MEF treatments are frequently employed to emphasize thermal and electrical effects, respectively. However, distinguishing between these effects within the same process can pose significant challenges. Employing the term MEF instead of OH might be more accurate due to the following reasons: (i) MEF involves more than just the heating aspect; and (ii) ohmic heating is simply an attendant effect of applying an MEF. In the scope of this review, the term MEF will be used preferentially applied where the implicit OH effect is considered.

### 1.2. Technologies–Status

Several technologies involve the direct application of electrical currents to food materials, each with specific mechanisms and applications in food processing and preservation. In addition to MEF, OH, and PEF, the literature commonly includes terms like Pulsed Ohmic Heating (POH) and High-Voltage Electrical Discharge (HVED). Figure 1 shows a schematic representation of different electric field-based technologies and their main features.

#### 1.2.1. PEF

PEF applications involve the application of high voltages in short pulses, typically in the nanosecond or microsecond range, with the primary aim of inducing electroporation of cell membranes. When cells are exposed to an external electric field, they develop a transmembrane potential. Once this potential surpasses a critical threshold, it triggers reversible or irreversible electropermeabilization of the membrane depending on the treatment intensity. PEF systems typically operate using square wave and alternating directional pulses. The technique’s efficacy hinges on operational parameters, with electric field (EF) strength, typically varying from 1 to 40 kV/cm, often cited as the most crucial factor [9,10]. Apart from EF strength, other factors like pulse number and duration, temperature, and product characteristics significantly influence the technique’s success and efficiency. The main applications include non-thermal inactivation of microbial cells and tissues softening to support the extraction of thermal labile biocompounds and induce textural changes.

#### 1.2.2. HVED

This technology operates on the principle of electrical breakdown in water, triggering both physical effects and chemical reactions, such as shock waves and ozone formation, respectively. This method involves the application of an electric field to create powerful electrical discharges or a plasma channel within a conductive substance entrapped between two electrodes. When compared to PEF, HVED causes greater damage to biological entities, affecting both cell walls and membranes, and is mainly used for extraction applications [11].

#### 1.2.3. MEF

MEF involves applying an electric field, usually ranging from 10 to 1000 V/cm, with an alternating current (AC) that periodically changes direction without specific time restrictions. In this technology, electric frequency plays a crucial role; at lower frequencies (typically between 50 and 60 Hz), electrochemical reactions can result in electrolysis, the creation of radical species and cause corrosion of the electrodes. However, employing frequencies in the range of kHz (typically between 10 and 20 kHz) or utilizing electrodes highly resistant to chemical reactions (like platinized titanium) can diminish or eliminate these electrochemical reactions. Depending on the electric field intensity and electrical conductivity of the sample, it is possible to generate and control the OH effect without a theoretical temperature limit. The main applications include continuous or batch thermal processing unit operations (e.g., high-temperature short-time pasteurization and sterilization of food materials, distillation, blanching, and hydrothermal extraction).

#### 1.2.4. POH

A key aspect for the development of new equipment and minimization of electrolytic effects was the appearance of Integrated Gate Bipolar Transistor (IGBT) power supply systems. IBGT systems allow one to modulate the amount of electrical power delivered to the material being heated, handling high power and voltage efficiently. According to Samaranayake et al. (2006) [12], pulse waveforms produced from an IGBT vary from those commonly created using high-frequency generators, and they can be altered separately by modifying parameters such as frequency, pulse duration, and duration between adjacent pulses, among others aspects. POH offers an opportunity to combine thermal and high-intensity electric fields.

To understand the global research status of these technologies, a bibliometric study of the literature published on the species was undertaken between 1987 and 2023 (December). Publications were retrieved from the Scopus database (Elsevier, Netherlands) using the following search strings: (i) (TITLE-ABS-KEY ({Ohmic Heating} AND {Food}); (ii) (TITLE-ABS-KEY ({Pulsed electric fields} AND {Food}); (iii) (TITLE-ABS-KEY ({Moderate Electric Fields} AND {Food}) (TITLE-ABS-KEY ({HVED} AND {Food}); (TITLE-ABS-KEY ({Pulsed Ohmic Hating} AND {Food}). The search identified a total 3146 publications, which were subjected to descriptive bibliometric analysis and mapping. Some quantitative descriptors can be found in Table 1. The number of publications regarding PEF and OH technology has been consistently growing since 1990.

Several key aspects, including technological advancements, the demand for innovative processing methods geared towards sustainability and health, and increased research efforts, contribute to the consistent growth in publications on OH and PEF technologies. Advancements in equipment, a deeper understanding of underlying principles, and improved control over these processes have rendered them more practical and attractive across diverse applications. These processing methods exhibit promising outcomes in preserving specific nutrients, enzymes, and flavors more effectively than conventional methods, owing to reduced or the absence of thermal load. Moreover, they contribute to the electrical inactivation of pathogens, enhancing product safety. The convergence of these factors has stimulated a steady growth in publications, highlighting the potential and ongoing exploration of PEF and OH technologies across various domains. Their versatility extends beyond food processing into fields such as pharmaceuticals, biotechnology, and materials science, broadening research interests. Additionally, subject areas like chemistry, immunology, microbiology, and medicine have also seen exploration in these technologies.

### 1.3. Industrial Applications

The accumulation of fundamental knowledge over the last few decades has furthered industrial applications. However, there is still limited information regarding the industrial facilities utilizing these technologies or the specific processed products available on the market processed by them. This is in part because of two reasons: (i) many companies do not publicly disclose the specific technologies to protect intellectual property, maintain a competitive edge, or simply because they do not consider it relevant information for public disclosure to prevent potential misconceptions among consumers; and (ii) these technologies are not widespread and are still used in niche or specialized applications that are not widely publicized or known outside of specific industries or research circles. Nevertheless, there is already a diverse range of suppliers offering OH and PEF industrial equipment, with a primary focus on food processing. This fact supports the growing demand and adoption of these technologies. Table 2 illustrates some of examples of industrial equipment for PEF and OH.

Most OH treatments are still centered on the thermal pasteurization of various sensitive food and beverage products, such as liquid eggs, fruit juices and pulps, and soups, among others. PEF applications leverage the electroporation mechanism to synergistically enhance the disintegration of cellular material and extraction yields of fruit juices, while preserving the fresh-like characteristics of the product. One of the most successful applications of PEF is in the initial frying stage of potato chip processing. This step softens the texture, facilitating improved slicing and enabling the creation of new shapes and cuts. Despite these industrial applications, the exploration of electrical processing in protein-based foods remains an underexplored area within food science research. This becomes even more critical when considering the increasing need to utilize alternative protein sources in the food industry, The intricate interactions between electrical processing and the structural, functional, and technological properties of proteins in food products presents a rich field for investigation. There is potential to modify protein structures and technological functionalities of protein fractions through electrical treatments. Comprehensive studies examining these interactions are notably scarce. Understanding how electrical processing influences protein conformation, aggregation, and functionality can support the development of innovative approaches in food technology, offering opportunities to tailor the texture, digestibility, and nutritional profiles of protein-based foods.

## 2. Food Proteins

Food proteins serve as macronutrients, supplying the necessary amino acids vital for human body growth and nutritional balance. Beyond their nutritional role, they serve as structural elements in food preparation, contributing to processes such as gelling, thickening, and emulsification. Additionally, their nutraceutical characteristics, including antioxidant and antimicrobial properties, confer physiological health benefits [13,14]. This is achieved through intricate physicochemical interactions with bioactive components, offering functional attributes that may contribute to disease prevention. Proteins are available from a variety of dietary sources, including animal- and plant-based diets, in addition to the prominent sports supplement sector [15].

### 2.1. Conventional Protein Sources

Animal-based proteins (from meat, poultry, fish, dairy, and eggs) have long been fundamental ingredients in the food industry. They hold immense importance due to their balanced nutritional profile, functional and technological properties, and consumer preferences. Figure 2 highlights some of these major sources, addressing their main biological properties as well.

(a)Dairy proteins

Bovine milk has long been a crucial protein source in the human diet, especially in infant nutrition. Milk protein ingredients also hold great interest not only because of their nutritional quality but also because of their specific technological functionality [16]. In bovine milk, approximately 80% of the total protein content is attributed to caseins, with the remaining 20% consisting of whey proteins [17,18]. In terms of available dairy ingredients, it is possible to find casein/caseinates, micellar casein, co-precipitates, milk protein concentrate, whey protein concentrates and isolates, and ultrafiltered retentate powder [19,20]. These protein ingredients possess good emulsifying properties and excellent water binding, thickening, and gelling properties. They are typically applied in infant formula, performance and health nutrition, nutritional bars, beverages, and processed yoghurt and cheese, among others [21,22]. In general, dairy protein ingredients have widespread use in beverages, confectionery, bakery products, meat and fish products, dietetic foods, infant formulas, and foods for the elderly, as well as specialty products catering to slimming, clinical and medical support, and sports nutrition [14,23]. Furthermore, milk proteins, such as casein, β-lactoglobulin (β-Lg), α-lactalbumin (α-La), bovine serum albumin, are used as model proteins in fundamental studies (e.g., accessing EF effects in proteins structure) and have several biotechnological applications.

(b) Meat Proteins

Meat is rich in protein, vitamins, minerals, and micronutrients that are crucial for the growth and development of the human body. The proteins in meat, constituting roughly 20% of a muscle’s weight, play a fundamental role in forming the structure of meat products [18]. From a nutritional perspective, the significance of meat lies in its high-quality protein, containing all essential amino acids, as well as its easily absorbable minerals and vitamins. Notably, meat is a valuable source of vitamin B12 and iron, which may be less readily available in vegetarian diets [24]. Muscle proteins can be categorized in three groups: myofibrillar, sarcoplasmic, and connective tissue proteins, composing about 50–55%, 30–34%, and 10–20% of the total protein in meat, respectively [25].

Collagen in particular, which is exclusive to the animal kingdom, plays structural and connective roles in various tissues such as the skin, bone, cartilage, tendon, and blood vessels. Partially hydrolyzed or heat-denatured commercial collagen forms gelatin, one of the most versatile meat protein ingredients, which is widely utilized as a food additive, including as a stabilizer, thickener, gelling agent, film former, whipping agent, or clarifying agent in various food products [26]. Another source of high-protein ingredients in processed meat and other food products is meat protein derived from lean tissue components or from by-products of meat processing [18].

(c) Seafood Proteins

Seafood represents an abundant and valuable protein source. The muscle of edible fish typically contains 16–21% protein, with slightly higher protein contents in fatty fish and crustaceans. In contrast to other animal-based protein-rich foods, consumers have long recognized fish as a high-protein food that is lower in energy and total fat, particularly saturated fat [18].

Structural proteins, which constitute about 70–80% of fish muscle, are soluble in cold, neutral salt solutions with a relatively high ionic strength. These proteins display significant functionality and are used in restructuring seafood products such as surimi. Various materials, including under-utilized species with low commercial value, can be employed in the production of surimi [14,18].

(d) Eggs Proteins

Egg proteins are acknowledged for their elevated nutritional quality, superb digestibility, and comprehensive supply of essential amino acids [14]. A whole egg is composed of 75% water, 12% protein, 12% lipids, and approximately 1% carbohydrates and minerals [21]. The food industry extensively utilizes egg products as a potent protein source, not only for their nutritional value and sensory attributes but primarily for their functional properties. This widespread incorporation into manufactured food products is attributed to the valuable contributions of egg proteins in various applications [18,27].

The separated egg white, egg yolk, and pasteurized whole egg can be processed into liquid, frozen, or powder forms and further used in the food industry because of their ability to foam, emulsify, gel, and thicken [28]. Their denaturation and coagulation at specific temperatures and the formation of a stable matrix upon coagulation is a beneficial functional characteristic of these proteins and has been explored over the years [29]. Egg proteins are sensitive to changes in conditions such as pH, thermal processing, and ionic strength, resulting in changes in their functionality [30]. For this reason, processing of these ingredients imposes challenges and often requires alternative and mild processing conditions.

(e) Wheat and cereal grains

Wheat provides the greatest amount of protein in the human diet of all plant sources. Everyone around the world eats bread, breakfast foods, pasta, and other basic products made of wheat [18]. Cereal grains and foods made from them have a protein content that ranges from 7 to 15%, which is typically less than foods containing animal protein on a dry matter basis. Grains are also used in the commercial manufacturing of protein ingredients. As a co-product of starch production, “vital wheat gluten” is extracted from wheat and added to a range of manufactured food products. Its protein content can reach 75–80% [31]. Gluten functions to enhance the protein content of products based on flour and also improves the water-binding capacity, such as in processed meat goods. However, a significant concern linked to the extensive use of gluten from wheat (as well as related proteins from barley and rye) is its association with celiac disease. This condition is characterized by inflammation of the small intestine, resulting from an inappropriate immune response to the prolamin family of storing seed proteins [14].

### 2.2. Dietary Transition—Alternative Sources

The global food industry is rapidly changing, with a notable shift from the aforementioned traditional protein sources to alternative ones in our diets. In recent years, consumer preferences have shifted towards alternative products, primarily led by health-conscious individuals seeking safer and healthier food choices [32]. Overconsumption of conventional proteins has been linked to heart disease, obesity, and certain cancers [33,34]. Shifting to alternative proteins can reduce health risks due to their lower saturated fat and cholesterol content, making them heart-healthy and decreasing the risk of cardiovascular disease, hypertension, diabetes, and overall mortality. Diversifying alternative protein sources in one’s diet also enhances overall nutrient intake, promoting a balanced and healthier diet [35,36]. Additionally, as the global population continues to grow, ensuring food security and environmental sustainability becomes increasingly challenging. Livestock agriculture contribute significantly to greenhouse gas emissions, deforestation, and water pollution. Therefore, shifting towards alternative protein sources contributes to balancing the current food system since their production requires fewer natural resources, produces fewer emissions, and has a lower ecological footprint [37,38]. In fact, recent life cycle assessment studies have revealed that plant-based protein products exhibit a reduced environmental footprint compared to traditional alternatives [39,40]. Ethical and religious considerations are also prominent issues concerning traditional protein sources. Conventional animal farming practices have faced criticism for their treatment of animals, which includes issues like cramped living conditions, overpopulation, and the use of antibiotics [41]. Alternative proteins offer a more humane approach to protein production by eliminating the need for raising and slaughtering animals, thus reducing animal suffering.

Alternative protein sources showcase diverse biological activities with associated health benefits and can be classified into several categories such as plant (legume and oilseed), algae, insect, and fungi/mushroom proteins, as presented in Figure 3.

#### 2.2.1. Plant Proteins

(a)Legume proteins

Proteins derived from legumes, including beans, chickpeas, lentils, lupines, peas, and mung beans, serve as a primary protein source and are rich in lysine and threonine [42]. Legume proteins exhibit functional properties such as emulsification, gel formation, and foam stabilization, which enhance their suitability as food ingredients [43].

Peas are recognized as a valuable alternative protein source due to their high protein content, particularly in the form of essential amino acids, including lysine and threonine. Moreover, peas have a low glycemic index, making them an attractive option for those concerned about blood sugar regulation. Numerous academic research studies have emphasized their anti-carcinogenic properties in contributing to the prevention of colon cancer, as well as their efficiency in treating leukemia, breast, pancreatic, prostate, and lung cancer [44]. Lentils are available in a variety of colors, each with their own nutritional profile and culinary diversity [45]. Their nutritional profile, including their rich content of B vitamins, minerals like calcium, phosphorus, and potassium, as well as oleic, linoleic, and palmitic acids, has been linked to potential health benefits in humans. These benefits encompass reducing cholesterol and lipid levels, as well as lowering the risk of colon cancer and type-2 diabetes [46]. Faba bean proteins display notable emulsion and foam stabilizing properties, although they may not match the effectiveness of soy protein isolate, which could be a limiting factor when considering them as an alternative protein source [47]. Nonetheless, it is feasible to enhance the functional attributes of legume proteins through various production and processing techniques. In fact, faba bean proteins have found success as alternative proteins in the creation of meat analogs using methods such as shear cell technology, wet spinning, and high moisture extrusion [48]. Lupin seeds are comparable in nutritional value to soybeans, particularly low-alkaloid varieties, with a high protein content up to 46% protein in some varieties. Besides their high content in functional compounds that contribute to health-promoting properties, they exhibit antioxidant and hypocholesterolemia activity, possess a low glycemic index, enhance mineral bioavailability, and have anti-allergic and anti-inflammatory effects [49,50]. Chickpea protein is a popular legume protein with good texture, capacity to bind to water and oil, and ability to form a gel. It can also stabilize emulsions and foams, such as soy protein isolate and whey proteins, which makes it a very interesting alternative protein. When compared to soy protein isolate, chickpea protein isolate absorbs more fat and a comparable amount of water [51]. Furthermore, studies show that chickpeas offer a significant advantage as an alternative protein ingredient as they positively influence the color acceptability of the products [47]. Mung bean proteins have also gained popularity due to their high protein content, low fat content, and a favorable amino acid profile. They are primarily composed of globular proteins, making them ideal for creating gels and stabilizing foams and emulsions, similar to chickpea and faba bean proteins [44]. This versatility allows mung bean proteins to contribute to the desirable textural properties of meat analogs while providing a balanced amino acid composition. Additionally, mung bean proteins have been documented to exhibit inhibitory effects on angiotensin-converting enzymes (ACE), along with antimicrobial and antifungal properties [52].

Despite their high protein content, legume proteins have low digestibility due to the presence of complex polysaccharides and oligosaccharides that are difficult to digest in the human stomach. In addition, compounds present in legume seed coats, such as tannins, polyphenols, and phytates, also inhibit protein digestibility, which can cause a prevalent human discomfort known as flatulence when consumed [53]. In this context, germination has been recognized as an efficient method to enhance the quality and nutritional prospects of legume protein. The germination process, commonly referred to as sprouting, entails soaking legumes in water and maintaining them in moist conditions until they initiate the germination phase [54]. Germination improves their digestibility, increases the amino acid content, and decreases the level of antinutrients [55]. Research conducted by Liu et al. (2020) demonstrated the efficacy of employing a hydration process followed by thermal processing to effectively reduce oligo-sugars and antinutrients in legume proteins, thereby diminishing the flatulence associated with their consumption [56].

(b) Oilseed proteins

In recent years, numerous oilseeds have gained recognition as alternative protein sources in the food industry. These include, among others, soybeans, rapeseed/canola, sunflower seeds, sesame, flaxseeds, pumpkin seeds, hemp seeds, chia seeds, and linseeds [57]. Soybean protein has already been widely used as an alternative source. Soybeans are a nutritional powerhouse, known for their rich protein content and essential amino acids. These proteins offer various health benefits, including anti-cholesterol, anti-hypertensive, anti-inflammatory, antimicrobial, antioxidant, and anti-cancer activities and their unique functional properties, such as emulsification and texturization, make them a valuable ingredient in various food products [58]. Flaxseed is abundant in bioactive components, including polyunsaturated fatty acids that promote cardiovascular health and lignans with antioxidant and anti-cancer attributes, and has notable foaming capabilities. Additionally, it showcases physiological benefits such as improved triglyceride and cholesterol levels, surpassing the well-known soy proteins in this regard [59]. Moreover, the reduced allergenic potential of pumpkin and hemp seeds [60], or the potentially non-allergenic nature of chia seeds, in contrast to legume proteins [57], provides an opportunity for their utilization as functional components in newly created food items. Rapeseed proteins have also garnered attention owing to their high solubility and foaming ability. Additionally, they exhibit a range of desirable attributes, including ACE inhibitory activity, antioxidant properties, bile acid-binding capacity, and anti-coagulation and anti-thrombotic potential [61]. While oilseed proteins offer numerous advantages, it is important to be aware of potential risks associated with certain plants like rapeseed. These plant proteins contain valuable nutrients for human consumption but also harbor toxic substances like erucic acid and sulfur compounds [62].

#### 2.2.2. Algae Proteins

Algae proteins, such as *Chlorella* sp. and *Arthrospira* sp. (commercially known as Spirulina), and several other marine microalgae are increasingly being recognized as alternative protein sources that offer a range of biological properties beneficial for both human health and environmental sustainability.

*Chlorella* sp. and *Arthrospira* sp. have garnered attention for their impressive nutrient profiles. *Chlorella* sp. is a green freshwater microalgae rich in essential amino acids, vitamins, and minerals, including vitamin B12, which is rare in plant-based foods [63]. *Arthrospira* sp., a blue-green cyanobacteria, is renowned for its high protein content, offering all of the essential amino acids [64]. Beyond their nutritional content, these cyanobacteria possess interesting antioxidant properties. Their high levels of chlorophyll, carotenoids, and phycocyanin make them effective in neutralizing harmful free radicals. These algae proteins are also rich in antioxidants that are essential in protecting the body from oxidative stress, reducing the risk of chronic diseases, promoting cardiovascular health, and supporting overall well-being [63]. Research is actively exploring specific health benefits associated with bioactive peptides sourced from spirulina. These peptides are being investigated for their antimicrobial, anti-allergic, anti-hypertensive, anti-tumor, and immunomodulatory properties [64]. In general, marine microalgae provide unique protein content while showcasing the benefits of sustainable aquaculture practices. These microalgae are not only rich in protein but also contain valuable omega-3 fatty acids, which are vital for heart and brain health. They also exhibit high antioxidant, anti-hypertensive, anti-coagulant, and immune-stimulant activities. In addition to offering an environmentally friendly protein source, their cultivation can minimize the strain on land resources and reduce greenhouse gas emissions associated with traditional livestock farming [65].

#### 2.2.3. Insect Proteins

Insect proteins, derived from species like crickets, mealworms, and grasshoppers, have gained attention as an eco-friendly and sustainable protein source [66]. The production of insect proteins requires significantly fewer resources, such as land, water, and feed, compared to traditional proteins, making them a compelling solution to address global food sustainability challenges [67,68].

One of the most intriguing aspects of insect proteins is their diverse biological activities. Several studies have indicated the presence of bioactive compounds and peptides in these proteins, leading to various health benefits. Insect proteins are rich in antimicrobial peptides that offer robust defense against bacterial, fungal, and viral infections. Moreover, these proteins demonstrate immunomodulatory characteristics, actively boosting and reinforcing the immune system [69]. Another noteworthy biological feature related to insect proteins is their capacity to act as antioxidants, which counteract oxidative stress and reduce the likelihood of chronic diseases. Additionally, bioactive peptides originating from insect proteins exhibit anti-inflammatory qualities, contributing significantly to the management of various health conditions [70]. Emerging research also suggests that certain insect proteins possess not only antiviral capacities but also anti-cancer properties, making them a potential resource in cancer therapy [70].

#### 2.2.4. Fungi/Mushroom Proteins

Fungi and mushrooms are rich in proteins, and their protein content varies depending on the species. These proteins are not only nutritionally valuable but also offer a broad spectrum of essential amino acids, making them suitable for human consumption. The amino acid profile of fungi/mushroom proteins often complements plant-based protein sources and represents an alternative to conventional proteins. In addition to proteins, fungi/mushrooms are excellent sources of dietary fiber, vitamins, and minerals, making them well-rounded nutritional options [71].

One of the most intriguing aspects of fungi/mushroom proteins is their immunomodulatory effects. Certain mushroom species, such as reishi, contain bioactive compounds that can stimulate and regulate the immune system. These compounds enhance the body’s ability to defend against infections and even exhibit anti-cancer properties by activating immune responses against cancer cells [72]. Fungi/mushroom proteins, such as shiitake and Agaricus, are replete with antioxidants, such as selenium. These antioxidants counteract oxidative stress, reducing the risk of chronic diseases and promoting overall well-being [71]. Furthermore, certain fungi/mushroom proteins contain natural antimicrobial and anti-inflammatory activities, such as those found in turkey tail mushrooms. These peptides not only protect the mushroom but also show potential in human medicine and the development of novel antibiotics [73]. Ultimately, multiple research studies have established that various mushroom species exhibit substantial inhibitory effects on diverse forms of cancer [74].

While the alternative proteins mentioned above have the potential to replace traditional protein sources, they frequently do not meet the industry’s functional and nutritional requirements [75]. To be a viable alternative, it is imperative to address the functional limitations associated with these novel proteins. Hence, gaining a deep understanding of the functional and technological properties of these proteins can aid in the development of processing methods that have the potential to modify and regulate protein functionality, ultimately enhancing their health-related advantages.

### 2.3. Functional and Technological Properties

The primary determinant of proteins’ distinctive characteristics lies in their amino acid sequence and the interactions established among them. These interactions formed within a protein and with other protein molecules are the foundation for both the structural and functional attributes of proteins [76]. Due to their particular functionality, proteins are used in food production with several technological purposes, acting as building blocks of food matrixes, forming and stabilizing interfacial systems, and protecting and transporting bioactives, among others. Furthermore, by manipulating protein structures and their aggregation, through processing techniques, a wide range of techno-functional properties can be improved or created [77].

A comprehensive investigation of protein structures can reveal many insights into their functionality, as the fundamental principles governing structure–function relationships are well established [78]. However, proteins in food products are also influenced by the complex composition of these matrices. This is particularly critical due the protein’s distinctive structural features, such as specific charge distributions, hydrophilic and hydrophobic regions, making them particularly prone to interact with various phases. Thus, the study of protein functional properties often resorts to a materials science approach and they are generally defined as solubility, gelling, foaming, emulsifying, water-holding, and fat binding capabilities.

#### 2.3.1. Solubility

Solubility refers to the protein’s capacity to dissolve in an aqueous solution. It is generally recognized that in their native state, proteins conceal non-polar and hydrophobic groups within their core, while hydrophilic groups tend to be on the surface [79]. Additionally to the positioning of polar and nonpolar/hydrophobic groups within the protein’s conformations, the surface charge plays a decisive role in protein solubility. The electrostatic repulsion, governed by the pH-dependent titration of surface charges, is also fundamental to define protein’s solubility [80]. At the isoelectric point, where protein molecules have no net charge, they are the least soluble. Under conditions where proteins bear a net positive or negative charge, the charged amino acid residues on the protein’s surface interact with the ionic groups in the solvent, promoting protein dispersion and solubilization. Well-described examples of the pH dependence of protein solubility are caseins’ isoelectric precipitation during cheese and yogurt production [81], isoelectric precipitations of proteins during purification [82], and pH shifts to improve solubility/extractability methods [83]. The ionic composition of the media also defines the interaction established between the protein and the solvent and other proteins. The presence of salts can increase the solubility (salting-in effect) by compensating the protein’s surface charge with oppositely charged ions, preventing the electrostatic interaction between proteins. Contrary to this, protein’s solubility can be decreased (salting-out effect), usually at high ionic strengths, once it can compete for the accessibility of water molecules [84]. Because of this, food processing conditions often need to be adjusted. For example, myofibrillar proteins display low solubility in the absence of salts, while having increasing solubility at higher salt concentrations—i.e., <1 M [85]. Thus, during the production of processed meats such sausages, salt is used to solubilize the myofibrillar proteins, enabling the formation of a consistent emulsified mixture, texturizing and stabilizing the food matrix. Another example is the cold gelation mechanism, where interactions between denatured proteins are promoted by adding salt or by a pH shift, triggering aggregation and gelation [86].

Protein solubility is also dependent on the native/denaturation state of the proteins, which, in turn, is dependent on the environmental conditions. Apart from the pH and ionic strength, factors such heat, electro-magnetic fields, mechanical action, and chemical agents, among others, can disturb the native structure of proteins and thus affect its solubility. The most common method used in food processing is thermal processing. Subjecting protein-rich food to heat induces physical modifications in protein structures, primarily involving unfolding and conformational shifts. These non-native proteins may consequently lose their solubility either by an increased exposition of hydrophobic residues or by the formation of protein aggregates [87]. Likewise, any other modification method that disturbs the protein native structure will have consequences on the amino acid positioning, net charge, and potential interactions, affecting protein solubility.

Solubility is one of the most fundamental functional properties of proteins, directly affecting other functional properties such as gelation, emulsifying, and foaming. Given the complexity of factor affection protein’s solubility and the potential impact on their techno-functional properties, this parameter must be carefully accessed and understood in order to attain the maximum techno-functional potential of each protein ingredient and protein-rich food.

#### 2.3.2. Interfacial Properties

Proteins display an amphiphilic nature due to the presence of both hydrophilic and hydrophobic amino acids. Because of this, proteins can effectively bind water and fat, defining their water binding capacity (WBC) and fat binding capacity (FBC). These capacities are expressed as the mass of water and oil absorbed per gram of protein and are associated with important properties such as flavor retention and texture (e.g., tenderness, juiciness, and mouthfeel) [88]. Furthermore, the interaction of proteins with water and fat influences their interfacial properties. The interfacial absorption can lead to structural rearrangements or more flexible or linear proteins (e.g., caseins), driving the system towards a new minimum in free energy and a subsequent reduction in surface tension [89]. In the case of more rigid proteins and protein bodies, the interfacial properties seem to be more linked to the orientation of polar and nonpolar groups toward the aqueous and non-aqueous phase [90]. In addition to the capacity of proteins to stabilize interfacial systems related to their surface-active properties, charge-related phenomena, as well as the capacity to form structured films between phases, are fundamental to explain foam and emulsifying capacities [89,91].

The ability to adsorb in interfaces is frequently harnessed for the formation and stabilization of multiphase foods, such as the dispersion of oil droplets in an aqueous medium or surrounding air bubbles, forming emulsions and foams, respectively. Protein foams are well described in the literature and food technology, such as whipped egg whites or their plant alternatives such as aquafaba. In such systems, mechanical action results in aeration, protein unfolding, and its re-orientation at the interface by polar mobility. The interaction, through electrostatic, hydrophobic, and hydrogen bonds, between proteins around the air bubbles results in a protective-layer film stabilizing the foam [90]. The most used method to evaluate the protein’s foaming ability is throughout the foaming capacity (FC), measuring the volume (%) of incorporated air after whipping, and foam stability (FS), analyzing the foam stabilization (volume) during a specific period [92].

The emulsifying capacity is also fundamental in technological applications of food proteins, allowing the formation of products such mayonnaise, ice cream, and spreadable and processed meat products. The mechanisms of emulsification are similar to the ones described for foaming, with the adsorption of proteins at the interface of two phases, the orientation of polar and nonpolar groups, and the decrease in the interfacial tension. Emulsifying properties are evaluated through the determination of the emulsifying activity index (EAI), measuring the quantity of emulsified oil per 1 g of protein, and the emulsifying stability index (ESI), verifying the emulsion’s resistance during a specific period [92].

As with all of protein’s functional properties, external factors affecting the protein structure and stability, such as pH, ionic strength, denaturation, hydrolysis, or aggregation, can modify the interfacial properties of proteins. For example, in one study, the preparation of legume protein extracts (i.e., chickpea, faba bean, lentil, and pea) by isoelectric precipitation or salt extraction resulted in different attributes. The extraction process caused differences in solubility, surface hydrophobicity and surface charge on the recovered proteins that consequently affected the emulsification capacity and stability [93]. In another work, the authors explored a partial hydrolysis of egg white proteins and its impact on their foaming properties [94]. The results indicated that partial hydrolysis changed the protein structure, exposed hydrophobic groups, promoted interactions, and increased protein mobility and adsorption at the interface. All of this resulted in the increase in the FC of about 40% and FS of about 20%.

#### 2.3.3. Network Formation

The ability of proteins to establish interactions and form supramolecular structures is a particularly important property for the food industry since protein aggregation and gelification play key roles in shaping the texture and structure of food products. In their native fold, protein tends to be stable in solution due to electrostatic repulsion, hydration, and entropic forces. In addition, the reactive amino acids, nonpolar and cysteine residues, are usually occluded inside the protein structure. Because of this, the aggregation process requires a driving force, such as physical modifications (e.g., heat, pressure, shear and electric fields), chemical modifications (e.g., pH, oxidation/reducing agents, organic solvents), or enzymatic action. After the native protein fold is disrupted, newly exposed reactive groups are free to interact and to form stable aggregates, and if the protein concentration is enough, the process propagates and forms a self-supporting gel [95]. Furthermore, the aggregation process of protein can be controlled through the composition of the media, pH, ions, and other biopolymers, and the processing conditions, resulting in an array of shapes, sizes, and affinities. In conditions where the electrostatic repulsion is strong, usually stranded aggregates are formed, often displaying fractal propagation. If the electrostatic repulsion is low, the aggregation process results in spherical aggregates [95]. In particular conditions, the aggregation process occurs in a more ordered way, and proteins can assemble into long fibers or tubes. Due to their diversity, tunability, and excellent functional properties, protein aggregates have been used in a range of techno-functional roles, including as thinkers, foam and emulsion stabilizers, fat replacement, and in the transport and delivery of bioactives, among others [96,97,98,99].

Closely related to protein aggregation, gelation is fundamental in structuring and texturizing food products (e.g., cheese, yogurts, jellies, puddings). Gelation of proteins in the food context usually occurs by three methods: the propagation of the aggregation process, if the protein concentration is above a certain value; the gelation of soluble aggregates induced by a change in the electrostatic environment (change in pH or ionic strength); or enzymatic crosslinking [100]. Changing the conditions under which the gel forms can yield diverse structural features in the resulting gels, giving rise to a variety of morphologies. These morphological characteristics encompass the thickness of the network strands, the size of the mesh, the viscosity of the entrapped phase, and the type and strength of the formed interactions [101]. The mentioned microstructural properties are the defining factors for the macroscale properties associated with gels. These include appearance (color, transparency), texture (firmness, force to cause fracture, elasticity), and water holding. All of these properties are significant to the organoleptic profile and stability of food products.

### 2.4. Emerging Challenges

Non-animal proteins have gained significant attention as viable substitutes for animal-based proteins following the growing interest in sustainable and healthy food alternatives. To develop nutritious and sustainable non-animal protein-based proteins, it is crucial to understand the diversity of protein sources derived from non-animal sources (including cereals, vegetables, pulses, algae, and fungi and bacteria) and their potential use as protein ingredients in food formulations. Besides this, advances in extraction and processing technologies should be made to maximize their potential [102].

In recent years, non-animal protein-based products have been introduced to the market, including meat substitutes, dairy-free beverages, and powders. Unfortunately, some non-animal protein-based products often display limited nutritional, sensory, and functional qualities over other products derived from meat and dairy. It is then of high importance to find strategies to overcome such limitations. For instance, by modifying their functional properties during processing or even by complementing them with other protein sources to meet the nutritional needs of humans [103]. The food industry has increased interest in non-animal proteins in their versatile forms (i.e., flour, concentrate, isolate, hydrolysate, or textured), specifically for their potential use as additives with specific functional properties that may enhance the technological features of food products or the main ingredient for developing meat analogues [104]. In fact, the latter have been trending upward among both vegetarian and non-vegetarian consumers, leading to a boost in the demand for these kinds of products and increasing the pressure for the food industry to present different options. Traditionally, meat analogues are made from plant-based proteins such as soy and wheat gluten and, more recently, pea protein [102]. Apart from these proteins, novel proteins sources such as algae and fungal proteins have been explored as binding, filling, and flavoring ingredients in the formulation of meat analogues. For instance, the incorporation of *Arthrospira platensis* biomass at several protein concentrations in a texturized soy base resulted in products with differentiated color and intense flavor. Meat-substitute production from fungal origin has been also explored [105]. Currently, the production of mycoprotein by an edible fungus (*Fusarium venenatum*) is the basis of Quorn^TM^ meat substitutes. Quorn^TM^ is an interesting food product that not only contains protein but also high quantities of fiber and starch, providing good textural and nutritional attributes to meat substitutes [106]. Dairy-free beverages are another example of animal protein substitution due to the progressive decline of milk consumption associated with lifestyle trends, lactose intolerance, allergic reasons, and health concerns related to animal-based products. Plant-based beverages are essentially derived from soy, almond, coconut, or rice. From the nutritional point of view, soy protein has a total protein content comparable to cow’s milk, containing all the essential amino acids [107]. Algae protein powders have also been gaining popularity for the enrichment of traditional food products such as pasta. Low amounts (below 3%) of *Dunaliella salina* powder were added in the preparation step of pasta to improve its nutritional value. Its incorporation enhanced water absorption, resulting in an increase in the pasta volume and weight, but also losses in cooking. The addition at 1% did not affect the flavor, mouthfeel, or overall acceptability, as shown by a sensory evaluation [102].

Aimed at reducing animal protein consumption, food research has also been focused on exploring the partial replacement of animal proteins with plant proteins in food formulations. Despite being considered a useful approach for tracking synergetic technological and functional behaviors of mixed protein systems, there is still an evident disparity related to the protein sources used in most mixed system studies. Indeed, dairy proteins are a frequently used animal protein source, while soybean and pea proteins stand out as plant sources. Therefore, future efforts should be made to cover distinct proteins sources from both origins. In addition, the behavior of these animal and plant protein mixed systems is frequently characterized during and/or after heat treatments [17]. In this context, emerging processing technologies with recognized potential to induce structural changes in proteins and impact protein functionality, such as the case of electric field processing technologies, should also be considered as a new viewpoint and opportunity for innovation. Electric field-based technologies may offer a competitive advantage by introducing phenomena such as electroporation and ohmic heating with effects at the macro, micro, and biological levels.

## 3. Electric Field Effects on Proteins

When a certain electric potential (V) is applied to a food sample with a particular conductivity, it leads to the flow of electric current. The ratio between the applied electrical potential and the distance between equipment electrodes (which defines volume chamber) is called the electric field and is often expressed as volt per centimeter or volt per meter. Electric field intensity is then dependent on the applied voltage output and distance between electrodes. Under applied electric fields, electric charges can move via either direct current (DC) or alternating current (AC). With DC, charges flow in a single direction and can be delivered in pulses by modulating or controlling the electrical signal to produce intermittent bursts or pulses of the direct current. In AC, charges cyclically change direction at regular intervals by using bipolar pulses or quadratic or sinusoidal waveforms. The frequency of electrical alternation between the positive and negative direction plays an important role once it dictates the putative occurrence of electrochemical events. A higher frequency of polarity on the electrodes effectively diminishes the buildup of charge on the electrode surface and mitigates electrochemical effects, thereby reducing redox (reduction and oxidation) reactions, electrolysis, electrode corrosion, and the formation of highly reactive molecules. When the frequency is low, the polarity reversal time is not sufficient enough to eliminate electrochemical reactions and simultaneous redox half reactions can occur.

Numerous research studies have highlighted the significant impact of electrical variables—such as electrical frequency and electric field—on protein structures. These variations can induce alterations in denaturation and aggregation pathways. It is becoming increasingly evident that these electrical effects should not be disregarded; instead, they represent an opportunity to fine-tune the properties of protein-based solutions by adjusting the applied electrical protocol. However, a significant challenge lies in disentangling these electrical effects from the thermal ones induced by the ohmic heating effect, for example. It is plausible that electrical processing merges electric, chemical, and thermal effects into a single treatment. Given the complex and dynamic nature of food proteins, the current fundamental research challenge relies around uncovering methods to precisely control electric processing parameters. This pursuit aims to achieve protein-based products with enhanced technological and functional properties, such as improved solubility, gelation, emulsification, and foaming characteristics. Table 3 highlights various effects of electric field-based technologies on protein-based foods.

### 3.1. Macrostructure Level

The unique operational principles related to electric field processing (i.e., thermal load and electrochemical effects) result in distinctive outputs in proteins and protein-based systems.

PEF has been effectively used to process several protein-based foods (e.g., milk and dairy, soy milk, and liquid egg) [128]. This technology is considered mild, arguably with limited action in food matrixes and macromolecular components, when compared with more aggressive technologies such as thermal processing. Yet, applied electric fields can cause cellular permeabilization due to the transmembrane potential. Following this, PEF can induce modifications in protein-rich food, such as muscle tissue, resulting in textural changes, increases in soluble protein, and increased digestibility. Some authors suggest that PEF application can increase proteolysis and physical alterations in the muscle structure, increasing tenderness and digestibility [129]. Other examples include the change in the organoleptic profile of cheese produced from PEF-treated milk [130] or the change in the physical and structural properties of processed liquid egg (microstructure, lipoprotein matrix, solubility, and on the mechanical properties of the resulting gels) [108]. PEF processing was used to process myofibrillar proteins extracted from pale, soft, exudative chicken breast. Different EF intensities (8, 18, and 28 kV·cm^−1^) were used, resulting in changes such as particle size distribution. Upon gelation of the myofibrillar protein solutions, these exhibited improved gel strength compared with the control sample (0 V·cm^−1^). Interestingly, the response to the electric field intensity was not linear and an optimum was found for the sample treated at 18 kV·cm^−1^ [109].

MEF technology, usually associated with OH, has a history of utilization in the food industry spanning over 30 years. This technology is not only used to process food but most recently has been seen as a method for modifying protein and protein-based structures. Naturally, the effects of MEF in proteins and protein-based foods are well reported, with examples of OH in the processing of meat or fish derivatives, soy, and dairy. The use of this technology often demonstrates changes in structures and organoleptic profiles, improving firmness, color, and water retention [113,114,131]. Recently, the effects of MEF (10 V·cm^−1^) in cooking chicken meat, testing variables such as wave type (square and sine) and electrical frequency (i.e., 50 Hz, 1000 Hz and 2000 Hz), against a conventional cooking process were reported. The different cooking protocols resulted in different cooking times, but the MEF cooked samples presented higher cooking yield and soluble protein content than the conventionally cooked samples. The authors also determined that the rheological properties of the cooked product were dependent on the wave type and frequency, increasing the structural strength with the increase in the electric frequency used, particularly for samples cooked under the square wave [115].

Certainly, the best described effect of MEF and attendant Joule or ohmic heating on dairy products comes from Caruggi, et al. (2019) [116], who studied the properties of acid milk gels produced with milk processed by MEF and convectional heating. The authors reported the influence of the voltage gradient, temperatures, and holding times on the final properties of the gel. The MEF-treated samples systematically displayed higher structural strength, lower syneresis, and produced a finer and more compact microstructure, improving the quality of acid milk gels. MEF effects have also been described in whey protein gels. Promoting the thermal gelation of WPI solutions with the Joule effect resulted in a gel with a lower structural strength and finer microstructure [117]. In a following study, Rodrigues et. al. (2021) [118] described the formation of WPI cold-set gels, where the preparation of the sol-gel was done by conventional heating and OH treatments at different MEF strengths (10 V·cm^−1^ and 20 V·cm^−1^) and electrical frequencies (50 Hz and 20 kHz). The samples processed by MEF were significantly different (*p* < 0.05) to the conventional ones is terms of sol-gel viscosity, gel strength, mechanical properties, swelling, and water-retention capacity. In addition, the MEF-treated gels displayed a finer and more homogenous microstructure and differentiate profile of network interaction, promoting more hydrophobic and electrostatic interactions, while reducing the disulfide bonding. The properties of the obtained gels were dependent on the electrical variables, with the use of higher electric field strength and lower frequency intensifying MEF effects. The application of OH treatments on the gelation of surimi is another well-documented example of MEF’s potential in processing protein-rich foods. The use of this technology enables the maximization of the functionality due to its fast and homogeneous internal heating. Other factors seem more related to the MEF effects, such as lower proteolytic activity and changes in disulfide crosslinking. Consequently, the gels prepared by MEF presented differentiated microstructural and mechanical properties in one study [113,132], contributing to the higher quality of surimi. MEF effects were also verified in gels of egg white and potato protein. Overall, the gels obtained under MEF demonstrated lower protein denaturation, lower water-holding capacity, lower rigidity, and higher flexibility [119,120]. Other authors have studied the effects of MEF in cheese emulsions, with treatments causing a weaker cheese structure, possibly associated with the potential removal of mineral content [133].

Overall, the electrical effects associated with PEF and MEF processing demonstrate the ability to affect the macrostructure of protein-rich food and protein-based food systems. Their effects are clearly dependent on the process variables such as electric field strength, wave shape, frequency, and number of pulses. Because of this, the reported effects are often distinct or inconsistent and should not be directly compared without considering the operational specificities of each report. Furthermore, foods are often complex in composition and structure, and MEF effects must be carefully evaluated and further explored in a comprehensive manner. Nonetheless, it is increasingly evident that MEF has the potential to impact the structural characteristics of macromolecules and cellular tissue, exerting a significant influence on structural integrity and network formation in protein food systems.

### 3.2. Nanostructure Level

The interest in protein-based nanostructures has grown during the past years. These structures can be intentionally crafted to enhance the encapsulation of a wide array of bioactive substances, showcasing intriguing properties and the potential to serve as a carrier system. For instance, they can reach particular targets that are difficult for their counterparts to access at the micro- and macroscale. Additionally, they can introduce novel textures and functional characteristics, making them promising options for delivering bioactive compounds intracellularly. Their superior surface area-to-volume ratio is crucial for in vivo applications. Furthermore, they boast a substantial capacity for loading active compounds, exhibit low density, and maintain high dispersion stability in aqueous environments. These protein-based structures can effectively function as carriers for bioactive compounds with varying molecular weights, prolonging the biological activity of substances in aqueous media. Depending on the intended application, diverse protein-based structures like gels, solid particles, and emulsions can be developed [78,134]. Frequently, in order to activate and regulate the functionality of proteins, enhancing their ability to interact with one another or other molecules, a motivating factor is necessary. This force is needed to alter the conformation of proteins and reveal their reactive groups, such as free thiols, hydrophobic regions, or charged species, which are typically concealed within the protein structure [135].

The field of food science and technology has extensively studied the interaction and aggregation of globular proteins. Nevertheless, the action of MEF can provide novel perspectives in the advancement of protein systems geared towards enhancing and managing functionality. MEF can change the natural equilibrium of interactions to change the structure of proteins. This, consequently, has a substantial effect on the protein’s ability to form organized structures, as well as its overall functionality.

The interaction and aggregation of globular proteins is a well-established and widely explored field of food science and technology. However, the MEF action can contribute to new insights on the development of protein systems aimed at improving and controlling functionality. MEF has the capacity to change protein structures by changing the natural balance of the interactions, which, in turn, can greatly influence both the protein’s functionality and ability to form structured systems [136]. An ongoing area of research involves the creation of protein fibrillar systems. Proteins sourced from various origins can be engineered in vitro to form 3D supportive structures, resulting in biomaterials with distinct morphologies and multifunctional properties. These systems serve as carriers for bioactive compounds in food formulations, drug delivery systems, and network systems for cell communication. A specific instance of such fibrillar systems is represented by amyloid fibrils, currently defined as “self-assembled and highly ordered peptide/protein aggregates associated with both disease and function” [137]. These fibrillar systems can be utilized to create protein models that aid in designing therapeutic compounds for treating currently incurable protein-related diseases like Alzheimer’s, Parkinson’s, and transmissible spongiform encephalopathies, all linked to alterations in protein conformation [137]. These fibrillar systems create a delicate and interconnected interface between food and health, offering significant potential for diverse applications. The mechanism of protein unfolding and aggregation, the size and hierarchical structure of the resulting protein and peptide fibrils (from the atomistic to mesoscopic length scales), the interactions between proteins, and the involved biological or artificial environments are the key events that cao and Mezzenga (2019) emphasized as being crucial to the final function of these fibrillar systems [138].

When the existing knowledge base is considered, it becomes evident that MEF has the potential to reveal novel perspectives in advancing these systems with the goal of enhancing their functionalities. One significant challenge in the development of these systems is achieving controlled aggregation during fibrillation and gaining a deeper understanding of how to promote organized formation over an amorphous shape [139]. MEF has been shown to notably enhance the surface hydrophobicity of β-Lg and modify the profile of free sulfhydryl (SH) groups by either disrupting disulphide bonds or revealing cysteine residues concealed within the protein structure. Environmental factors like pH, ionic strength, and temperature also had an impact on these modifications [126]. Furthermore, the combination of internal heating and moderate-intensity EF can boost protein hydrolysis, a process frequently linked to the yield of fibril formation [140]. According to Pereira et al. (2016) [122], MEF treatments at 12 V/cm led to a 30% decrease in free thiol groups, a size reduction in the produced aggregates from 300 nm in the conventional procedure to 120 nm under MEF, and alterations in morphology that promoted the creation of lengthened structures.

Subaşı et al. (2021) [123] documented the impact of MEF on the treatment of caseinate and sunflower protein. MEF influenced the secondary and tertiary structures, as well as the thermal properties of protein systems. Additionally, the particle size and homogeneity were decreased, and the interfacial tension at the air/water interface was altered as a result of the MEF activity. The capacity of MEF to impact whey protein aggregation and functionality during heating was demonstrated through the application of a cold gelation approach. In this sequential procedure, protein denaturation was first initiated through conventional heating and MEF methods using the OH effect. Subsequently, Fe^2+^ was introduced at room temperature, leading to additional aggregation [141]. The inclusion of MEF (10 V/cm) to allow OH of samples resulted in a decrease in particle size distribution and induced alterations in the physical stability, rheological behavior, and microstructure of the generated microgels. Moreover, there was an opportunity to enhance the inclusion of Fe^2+^ and adjust the rheological characteristics of the gels. Likewise, the application of MEF to lactoferrin led to distinct physicochemical properties, including a decrease in aggregate size, polydispersity, free SH content, and surface hydrophobicity [127]. The creation of emulsions from heat-treated lactoferrin, using both MEF and conventional heating, yielded a gel-like substance with distinct properties. The MEF-treated samples exhibited a finer and less rigid microstructure, displaying more fluid behavior.

Currently, strong evidence can be found in the literature on the effects of the electric field on the process of aggregation/denaturation of proteins to produce nanostructures, which could help to open up new horizons for interesting applications in the food industry.

Using proteins for the encapsulation and controlled release of bioactive compounds appears to be among the most compelling applications. However, overcoming challenges related to scaling-up production in a competitive approach is a critical consideration. Additionally, there is a need for a more comprehensive understanding of how the produced structures interact with the human body. Addressing this involves conducting toxicological studies to confirm and ensure their safety.

### 3.3. Functional

The research on the potential of electric field processing to replace conventional thermal processing is currently well established not only to improve the shelf life of food products (through microbial inactivation) but also their quality. Despite such promising applications, a distinct approach has been gaining the interest of food researchers: the electric field application to change the inherent structure of food proteins [142]. In fact, currently, it is possible to find several studies where electric fields are applied on both animal and plant proteins to change their structure, giving rise to the formation of novel food ingredients such as gels or films with tailored functionality according to the intended food application [118,125].

The application of MEF and its attendant OH effect on dairy proteins has been comprehensively studied over the last few years, with demonstrated promising results in modifying their denaturation and aggregation pathways. For instance, the denaturation levels of whey protein isolate (WPI) solutions were modified in the presence of an electric field. A decrease in the percentage of free sulfhydryl groups, reduction in the protein aggregates size, and change in their morphology to a fibrillar shape was observed. This led to the formation of protein aggregates with distinctive features with recognized potential to form physical gels that could be used in several food applications [122]. The production of WPI cold-set gels upon OH treatment mediated by iron addition promoted the formation of hydrogels with more uniform and compact fine-stranded microstructures [141]. The development of hydrogels or emulsions from globular whey proteins to be used as food thickeners or as delivery systems of functional biomolecules or bioactives (such as heat sensitive ingredients) has also been achieved through pre-treatment with MEF [143]. Interestingly, it was recently demonstrated that the control of MEF variables allows one to control the functionality of whey proteins gels [118]. Indeed, this control not only has an impact on protein denaturation and aggregation pathways but also allows one to control the gelation process (i.e., it induces molecular interactions, influences the protein network formation, and establishes the final gel properties). Additionally, the conjugation of high electric field strength and low frequency has led to the production of weaker gels which are more elastic and possess higher water retention and swelling capacities. The formation of films from OH-treated WPI solutions was also studied. The attainment of thinner films, possessing lower permeability to water vapor but with similar mechanical properties to conventional ones, was also reported [144]. Apart from the reported studies on whey proteins, other dairy proteins such as casein have been submitted to MEF [121]. Sodium caseinate solutions submitted to MEF treatment resulted in gels with distinguished features, namely with lower water-holding capacity compared to the unheated sample. In one distinct study, the emulsifying and film-forming characteristics of colloidal dispersions of caseinate submitted to MEF treatments were assessed and compared with a control sample (not submitted to OH). The MEF-treated dispersions displayed a smaller particle size and developed emulsions with higher physical stability. The assessment of the mechanical properties of the produced films validated their higher mechanical strength and extensibility [145]. The impact of MEF on the heat-induced gelation process of other animal proteins such as egg white proteins has also been assessed. The gels formed presented a more open and porous structure compared to those obtained under a conventional heating treatment [119]. The interest in plant proteins as an alternative to animal proteins has attracted increasing attention from consumers due to their lower cost, better environmental sustainability and energy efficiency. Therefore, it is not surprising that studies focusing on plant protein functionalization are currently increasing. For instance, the application of MEF to induce protein–lipid films formed by heating soybean milk was studied. Films displaying better yield, film formation rate, protein incorporation efficiency, and rehydration capacity were produced [146]. More recently, it was demonstrated that MEF treatments at 4 and 6 V.cm−1 improved the gelation capacity of soy protein gels. The application of OH treatments and consequent impact of the inherent MEF on the gelling properties of pea protein was also studied. Chen and et al. (2022) [125] have demonstrated that OH-treated pea protein gels displayed the highest water-holding capacity and a uniform three-dimensional network structure when compared with conventional heating-induced gels. Thermally induced gelation of papatin-enriched potato protein was studied under the influence of MEF. MEF-treated potato protein gels possessed lower water-holding capacity than those obtained under conventional thermal treatment. Furthermore, these gels displayed a more gap-like structure, as well as frayed areas within the network [120].

PEF processing has also shown the potential to modify the inherent structure of proteins. Indeed, the growing number of reports on induced changes in proteins, especially in enzymes, has opened the path for focusing the research on proteins’ structural modification under PEF processing. However, little research has been conducted to explore the modification of molecular structures, or even the induced changes in denaturation/aggregation pathways and, thus, in the functional properties of PEF-modified protein-based systems [142]. For instance, the enhancement in egg white proteins and β-Lg gelation after the application of high-intensity PEF with long-length pulses was assessed. Protein gelation was improved due to an apparent synergetic effect resulting from the combination of a heating phase and PEF treatment. In addition, the gelation behavior of both proteins was proved to be affected by PEF treatment. The gelation rate of β-Lg was improved, while egg white proteins suffered a partial decrease in the rate of gelation [110]. Even though some studies focused on the structural modification of whey proteins and their fractions upon PEF application, information regarding the mechanisms of action on their physicochemical properties is limited. Consequently, the impact on the functionalization of such proteins is far from being understood. PEF processing is also capable of modifying the secondary and tertiary structure of plant proteins. Indeed, the impact of PEF in the structure and physicochemical attributes of soy protein isolate was also evaluated. Modifications in its solubility were observed upon PEF application and it was clear that an increase in PEF intensity and processing time was responsible for a decline in solubility, caused by protein denaturation and consequent aggregation [111]. Zhang et al. (2017) reported that PEF treatment of canola seeds increased the solubility, emulsifying, and foaming properties of the extracted protein [112]. Recently, a combination of PEF and pH shifting has efficiently improved the solubility of a commercial soy protein isolate. Moreover, functional properties like emulsifying and foaming were drastically enhanced by combining PEF and pH-shifting treatments [147].

Currently, it is possible to find strong evidence in the literature concerning the electric field effects on proteins’ functional properties. Yet, there is still a lack of detailed information related to their mechanics of action, particularly regarding PEF processing. Therefore, further studies are needed to better explore these effects and fully discover the potential of electric field processing to tune protein networks.

### 3.4. Biological

Several studies have explored the benefits of electric field processing in various areas, indirectly resulting in different biological outcomes. Some examples are the extraction of bioactive compounds (with antioxidant and antimicrobial properties), enhancements in food processing through changes in enzyme activity, improved microbial inactivation, and the development of enhanced food products derived from the byproducts of the food industry, offering potential beneficial health effects [148,149].

The extraction of bioactive compounds from various plant-based products has gained considerable importance through their possible biomedical and technological applications, such as the development of pharmaceutical and food products [150]. The conventional extraction methods employed involve leaching, as well as the application of solvents that can prove environmentally challenging, as well as not being time efficient or energy efficient [151]. PEF is recognized by its electroporation effects that support extraction strategies. In addition, some observations have been made regarding the efficiency of MEF in the extraction of bioactive compounds. This is due to the electric field promoting moisture evaporation in the plant matrix, causing considerable pressure to build up in the cell walls, eventually leading to its rupture and the release of the contents inside, including the desired bioactive compounds [151]. Several studies already seek to explore the potential of this interaction, such as extracting polyphenols and carotenoid compounds from tomato byproducts without the use of organic solvents whilst utilizing this method [152]. MEF-assisted extraction has been shown to be up to 63% more energy efficient than conventional extraction methods and yields extracts containing considerable quantities of bioactive compounds, as well as high antioxidant activity [153], further establishing the effectiveness of MEF as an extraction method for these compounds. However, there is still a need to comprehend the proper balance between achieving higher extraction efficiency and preventing the degradation of target compounds due to the OH effect.

Regarding the safety of developed food products, MEF presents effective interactions when dealing with pathogens, mainly due to offering both thermal effects (due to Joule heating), which induce irreversible damage to cellular membranes [153], as well as non-thermal-induced electroporation, leading to the leakage of cell contents [154]. The combination of these effects proves to be highly beneficial in food processing, particularly when contrasted with conventional heating methods that lack the non-thermal effects offered by MEF. As a result, conventional heating treatments often result in less microbial inactivation compared to MEF treatment [155]. The presence of some enzymes in foods can also be detrimental to the overall quality of the food product, with changes in flavor, aroma, and nutrient composition [156]. Mannozzi et al. (2019) [157] conducted MEF and PEF treatments in both carrot and apple juice samples, with successful inactivation of peroxidase in both juices and polyphenol oxidase in apple juice, whilst increasing antioxidant activity and retention of plant secondary metabolites. Samaranayake & Sastry (2016) conducted MEF treatments on tomato samples with noticeable non-thermal effects on pectin methylesterase inactivity, with a higher electric field strength resulting in a decrease in the activity of this enzyme [158]. Overall, both non-thermal and thermal effects of MEF treatments prove to be useful in food processing through microbial eradication, as well as increasing the quality of food through enzyme inactivation and retainment of desired properties in food products. In addition, MEF seems to possibly impact enzyme activity in both a negative and positive manner. Previous studies have recorded an increase in the enzyme activity of several enzymes, such as α-amylase samples when exposed to a frequency range from 1 to 60 Hz [159], as well as pectinase activity in wine must samples [160]. These increases in enzyme activity through MEF are beneficial since both of these enzymes have important roles in enhancing the quality of several food products. This further underlines the potential use of MEF as an interesting and sustainable method to be used in food processing.

MEF use has shown potential in immunological studies. Food allergies are a public health complication that appears to be worsening in past years. Not only is food allergy prevalence expected to increase globally in future years, but also children with food allergies appear to be more likely to maintain this health condition into late-childhood or even teenage years [161]. The most relevant and common food allergy in infants is cow’s milk allergy, presenting a prevalence of 5% in children [162]. Cow’s milk allergy can be associated with different allergenic molecules, with Bos d 4 and Bos d 5, corresponding to α-La and β-Lg, respectively, being the main sensitization allergens [163]. In that regard, the effects of MEF in the thermal denaturation/aggregation process of these proteins could exhibit a relevant impact on the allergenicity of milk proteins since denaturation causes conformational changes in tertiary structures of proteins, whilst enabling interactions with other molecules, which can change the sensitization potential of these proteins in allergic individuals [164]. Additionally, the non-thermal effects of MEF can prove beneficial in these studies since the reduced rate of heating can attenuate the formation of neo allergenic compounds that are more prevalent in conventional heating methods through Maillard reactions [165]. There have been attempts at understanding and exploring the potential of MEF in altering the immunoreactivity of cow’s milk proteins, mainly β-Lg. These studies concluded that the presence of an electrical field during OH treatments has an effect on β-Lg immunoreactivity. However, more studies are necessary for a better understanding of this fact. This is because the electric field strength and time of heating can either increase or decrease immunoreactivity, meaning that standard work parameters need to be established with this method to achieve satisfactory results [166]. Nevertheless, it is clear that the non-thermal effects of OH treatments can be explored further in the immunological field. The immunoreactivity of other common allergens, such as soybean trypsin inhibitor, has also been shown to be positively affected by MEF treatment [167].

### 3.5. EF Mechanisms

In this review we have summarized the effects of electrical field-based technologies on protein properties and in the food matrix. Despite the multitude of studies conducted, the underlying mechanisms of these modifications remain incompletely understood. This knowledge gap stems from the diverse application protocols associated with EF-based technologies, which exhibit great versatility. The delivery of EF to a given matrix can vary significantly based on parameters such as the intensity of the applied EF, time scale of application, occurrence of heat dissipation, EF-induced electrolysis, and type of electrical wave applied, among others. The interaction between electricity and molecules is inherently dependent on other factors such as ionic strength, pH, and media composition. Drawing upon the current state of the art, some observed EF effects appear to be linked to changes in amino acid and peptide interactions, disruptions in the mobility of specific groups, and the disturbance or reorientation of secondary and tertiary protein structures. It is evident that the nature and extent of the effects associated with EF is vast and often specific to the particular technique of the target matrix. An EF of higher intensity, often associated with PEF and HVED, is reported to cause membrane permeabilization and even cellular rupture. This naturally has an impact on the structural integrity, texture, and digestibility of protein-based foods such meat or other cellular tissue. Furthermore, the EF action leads to strong polarization, alignment of the proteins with the field, and disruption of intermolecular bonds, causing conformational changes, denaturation, and aggregation. When it comes to EFs of lower intensity, often the effects are not so clear due the lower magnitude and the occurrence of parallel phenomes such as heat dissipation. However, several studies point to the existence of EF effects even at residual EF intensities. The alternation of the field direction, associated with AC, results in energy dissipation through molecular motion, causing changes in the protein’s structure. Other factors such as an increase in the EF strength and the frequencies used are decisive in the EF’s action mechanisms, with the field strength dictating the polarization of the molecules and the frequency dictating the amplitude of motion. Furthermore, there is evidence of a cumulative effect in EF action, where prolonged exposure to EFs yields modifications of higher magnitude, regardless of the EF intensities of frequencies used.

Further research is necessary to fully elucidate the EF effects on protein molecules and complex food matrices. However, the recognized potential of EF-based technologies and the inherent EF effects plays a pivotal role in the advancement of novel food processing techniques and products. These innovations are essential contributors to the ongoing transformation in the paradigm of food production and consumption, with a specific emphasis on emerging proteins.

## 4. Conclusions

The need for a dietary transition to other protein sources like plant-based, microbial, or insect sources presents numerous technological challenges regarding their functional properties such as solubility, gelation, and emulsion, among others. There is a need for a more comprehensive understanding of the functional and biological properties of these alternative protein sources under food processing methods. Electric field processing brings new processing variants such as reduced thermal load, Joule heating, and electrochemical effects that seem to interact and change the structural behavior of food proteins. However, it is not yet possible to explain the general mechanisms that support these alterations and whether they can be explained by electrostatic perturbations, chemical effects, or heating kinetics, for example. A case-by-case analysis has been adopted, benchmarking conventional methods, in an attempt to distinguish between thermal and non-thermal effects promoted by various electric field-based technologies. However, numerous variables can influence these responses, such as non-measurable internal heating, as well as the interplay with intrinsic factors related to the physical and chemical composition of samples. Proteins are dynamic entities that are influenced by the physical and chemical environment, wherein factors like pH, ionic strength, and the presence of molecular ligands can significantly impact their structure and behavior. It is apparent that fundamental research on this topic is still in its early stages, highlighting the need for additional knowledge to design electric field processing for enhancing protein functionality. These technologies hold significant promise in developing novel functional protein-based foods, alongside potential applications in biomedical and pharmaceutical fields, potentially supplanting the use of chemical or enzymatic methods. With the emergence of alternative protein sources in the food industry, these technologies can assist in extracting protein-rich fractions from plant and microbial biomass. Additionally, they enable the control of their functionality, thereby enhancing their health-related benefits. They potentially offer a sustainable approach to pasteurization and sterilization unit operations in the food industry by increasing process efficiency, utilizing renewable energy sources, and reducing water consumption (no need for boilers or heat transportation). OH and PEF, historically regarded as sister technologies, are currently converging by fine-tuning electric protocols to combine thermal and electric effects into a single treatment, while maintaining acceptable energy inputs.

## Figures and Tables

**Figure 1 foods-13-00577-f001:**
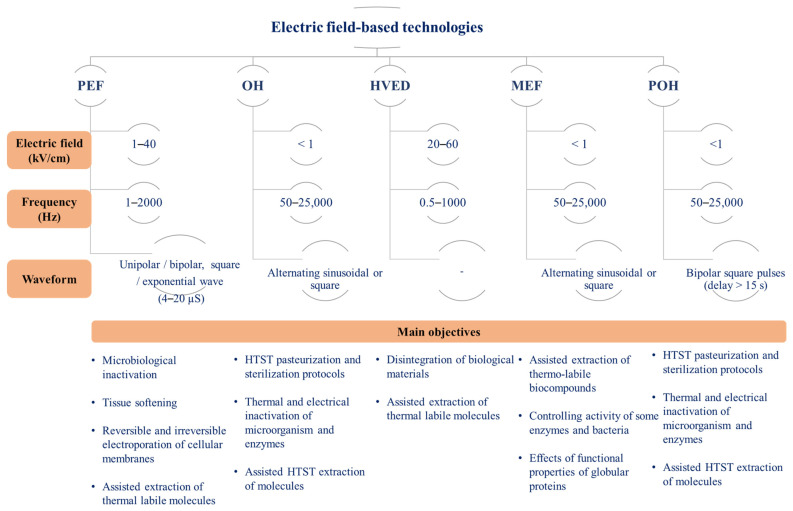
Electric field-based technologies and main processing objectives.

**Figure 2 foods-13-00577-f002:**
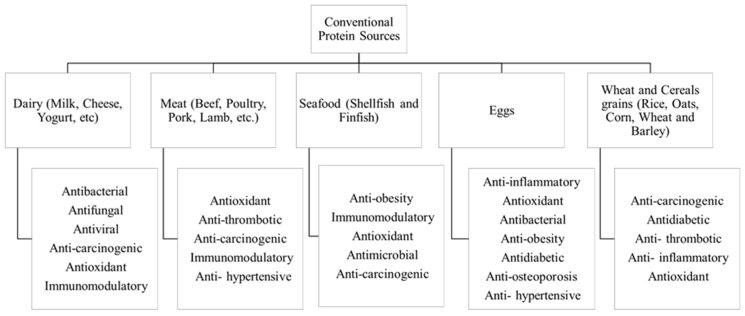
Conventional protein sources and associated biological activities.

**Figure 3 foods-13-00577-f003:**
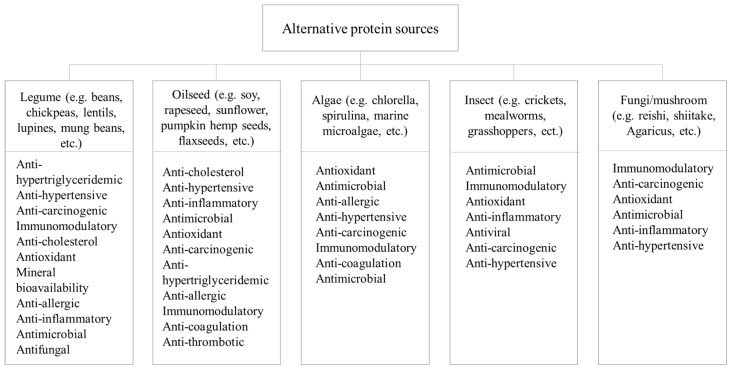
Alternative protein sources and associated biological activities.

**Table 1 foods-13-00577-t001:** Bibliometric survey of scientific research-related electric field technologies.

Technologies	Total Documents	Article	Review	Main Authors	Main Subject Areas
PEF	2256	1113	485	Martin-Belloso, O.Vorobiev, E.; and Barba, F. J.	Agricultural and Biological Sciences; Engineering and Chemical Engineering
OH	779	469	111	Sastry S. K; Pereira, R. N; and Vicente, A.A.	Agricultural and Biological Sciences; Engineering and Chemical Engineering
MEF	76	49	16	Pereira, R. N; Vicente, A.A; and Sastry, S. K.	Agricultural and Biological Sciences; Biochemistry
HVED	21	13	4	Vorobiev, E.; Barba, F. J.; and Babic, J.	Agricultural and Biological Sciences; Engineering and Chemical Engineering
POH	14	10	3	Kang, D.H.; Kim, S.S.; and Pereira, R. N.	Agricultural and Biological Sciences; Engineering and Chemical Engineering

**Table 2 foods-13-00577-t002:** Examples of companies providing industrial OH and PEF equipment or tailored solutions. All URL accessed on 31 December 2023.

Technology	Company	Country	Applications
OH	Raztek http://raztek.com/home.html	USA	Liquid egg and egg white
OH	JBT Corporate https://www.jbtc.com/	USA	Liquid, semi-liquid, high-viscosity products containing fibers, small cells; puree, soups, sauces, fruit preparations and fruit jam with dices
OH	Emmepiemme https://www.emmepiemme-srl.com/	Italy	Fruits and derivatives, vegetables, dairy products, egg products, algae, syrups, sauces, and ready-to-eat dishes
OH	C-Tech Innovation https://www.ctechinnovation.com/	U.K.	Custom-built units for food and beverage pasteurization and sterilization
PEF	DIL/ELEA https://elea-technology.com/	Germany	Large-scale units for different applications such as fruit beverages, wine, vegetable preparations, and solid foods
PEF	Energy Pulse Systems https://energypulsesystems.pt/eps/	Portugal	Custom-built units for inactivation of contaminants and extraction (e.g., juices, wine, microalgae)
PEF	Pulsemaster https://www.pulsemaster.us/	The Netherlands	Food preservation and process improvement for food and beverage industry
PEF	Opticept https://www.opticept.se/	Sweden	Extraction and preservation of beverages (e.g., olive oil, fruit juices, and wine)

**Table 3 foods-13-00577-t003:** Examples of main effects of PEF and MEF in the structural and functional properties of proteins and protein-based foods.

Technology	Protein Fraction	Operational Conditions	Effects Observed	Reference
PEF	Liquid whole egg	19–37 KV·cm^−1^, 18–30 µs	Increase in water-holding capacity, protein stability loss, changes in microstructure, and increase in hardness in heat-induced gels	[108]
	Chicken myofibrillar protein	8, 18, and 28 KV·cm^−1^	Decrease in particle size, increase in gel strength	[109]
	Egg white proteins and β-lactoglobulin	12.5 kV·cm^−1^, 1–10 pulses of 2–2.3 ms at	Partial denaturation, changes in denaturation temperature, formation of covalent aggregates	[110]
	Soy protein isolate	0–30 kV·cm^−1^, square bipolar pulses, 2 μs to 547 μs	Changes in secondary and tertiary structure, changes in disulfide bonds, and exposure of hydrophobic residues	[111]
	Canola protein	35 kV, 185.84 s of residence time, 806.75 Hz of pulse frequency, and 7.88 μs of pulse width	Increased functional properties (solubility, water-holding capacity, oil-holding capacity, emulsifying capacity and stability, foaming capacity, and stability), changes in secondary structure, changes in SH content, and increased hydrophobicity	[112]
MEF	Surimi	6.7–16.7 V·cm^−1^, 90 °C, 10 kHz	Increase gel strength, reduced free SH content	[113]
	Pork meat ball	20 V·cm^−1^, 50 Hz	Higher yield strength, more uniform microstructure	[114]
	Chicken meat	10 V·cm^−1^, 50, 1000 and 2000 Hz, sine and square wave	Increased cooking yield, higher water-soluble protein, changes in rheological properties, and increased cell disintegration	[115]
	Acid milk gels	25, 40, 55 V·cm^−1^, 60 Hz, sine wave	Gels with higher structural strength, lower syneresis, and finer and more compact microstructure	[116]
	Whey protein isolate gels	4–22 V·cm^−1^, 25 kHz, sine wave	Lower free SH content, lower protein denaturation, smaller particle size, lower gel strength	[117]
	Whey protein cold-set gels	10 and 20 V·cm^−1^, 50 and 20 kHz, sine wave	Lower free SH content, lower particle size and PdI, lower consistency index, more compact and homogeneous microstructure, softer and more elastic gels, higher swelling and water retention capacity.	[118]
	Egg white gels	4.3–17.2 V·cm^−1^, 10 kHz, rectangular pulses	Lower protein denaturation, lower water-holding capacity, increased gel firmness, lower hydrophobic interactions	[119]
	Potato protein isolate gels	2.5–24 V·cm^−1^, 10 kHz, rectangular pulses	Lower protein denaturation, lower water-holding capacity, increased gel firmness, lower hydrophobic interactions	[120]
	Sodium caseinate gels	2–17 V·cm^−1^, sine wave	Reduces aggregation, increasing protein solubility, gels with lower values of strain at rupture and water-holding capacity	[121]
	Whey protein aggregates	6 and 12 V·cm^−1^, 25 kHz, sine wave	Reduction in free SH, solubility increase, reduction in particle size, morphological changes	[122]
	Sunflower protein and sodium caseinate	10 and 150 V·cm^−1^, 60 Hz, sine wave	Changes in secondary and tertiary structure, decrease in denaturation temperature.	[123]
	Cold gel-like emulsions of lactoferrin	9–22 V·cm^−1^, sine wave	Changes in secondary and tertiary structure, reduction in particle size, gel-like emulsions with a less rigid structure	[124]
	Pea protein isolate	5, 10, and 20 V·cm^−1^, 50 Hz and 20 kHz	Reduction in particle size, changes in secondary and tertiary structure, reduction in SH content, increase in surface hydrophobicity, increase in water-holding capacity, and reduction in gel strength	[125]
	Β-lactoglobulin	20–80 V·cm^−1^, 20 kHz, sine wave	Changes in denaturation profile, changes in secondary and tertiary structure, increase in surface hydrophobicity	[126]
	Soybean protein isolate	4–10 V·cm^−1^, 50 Hz, sine wave	MEF promotes protein unfolding and exposure of hydrophobic residues, improves aggregation and gelation.	[127]

## Data Availability

The original contributions presented in the study are included in the article, further in-quiries can be directed to the corresponding author/s.

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
