# Peer review of "Electrical Fields in the Processing of Protein-Based Foods"

_foods, 2024, doi:10.3390/foods13040577_

Round 1

Reviewer 1 Report

Comments and Suggestions for Authors

This study reviews the species of electric field processing and industrial applications; the species of food proteins including conventional protein sources, dietary transition-alternative protein sources like plant proteins, algae proteins, insect proteins, and fungi/mushroom protein; functional and technological properties of food proteins; electric field effects on proteins including macrostructure level, nanostructure level, functional, and biological properties. The review is very interesting to the research and food industry processing. Suggestions,

1)     Line 159, in figure 1, the words are too small to read;

2)     Line 16, 97,126,1082,1084,1092, 1094, 1117, and 1126, the authors refer to non-thermal effect or non-thermal technology. Now the cold plasma is a non-thermal food processing technology and has many publications in recent 15 years, and the effects of cold plasma on food proteins have many articles. The authors should give the difference between electrical field based technologies and cold plasma in this reviewing paper.

3)     In table 3, SH and PdI should be noted.

4)     In many places, the authors used OH (ohmic heating), please use OH technology or OH treatment.

5)     Although the authors summed up the effect of electrical field based technologies on protein properties in food matrix. The authors had better give a theory or a hypothesis for explaining how electrical field based technologies act on the proteins in food matrix, so that the following researchers can find evidences or supporting data.

Author Response

We acknowledge the comments of the Reviewer 1 on the manuscript foods-2827923, which helped us to improve its overall quality. 

Authors comments can be found in the attachment. 

Reviewer 2 Report

Comments and Suggestions for Authors

Foods-2827923 Electrical Fields in the Processing of Protein-Based Foods

This is excellent manuscript reviewing the modern method of electrical field application in effort to replace animal proteins with vegetable ones. I have some general comments and specific comments.

General comments

-          Main objective of the manuscript is explained by the sentence “Main objective of this review is to give a comprehensive overview about the electric field processing of protein-based foods covering both fundamental and applied aspects of protein science. In connection with this, it would be desirable to include the work as one main citation Willett W, Rockström J, Loken B, et al. Food in the Anthropocene: the EAT–Lancet Commission on healthy diets from sustainable food systems. Lancet 2019; published online Jan 16. https://doi.org/10.1016/S0140-6736(18)31788-4

Your manuscript is full of methods how to change protein of animal origin by proteins of plant origin. Lancet paper is focused on this task for humans.

-          Part a) Legume proteins is excellently written but we have generally problems how to teach humans to eat them. There is flatulence problem. This problem can be fixed by using germinated legumes. Germination breaks down higher carbohydrates that are difficult to digest. Czech version of the cookbook can be find at https://www.vyzivaspol.cz/nova-e-publikace-klicim-klicis-klicime-aneb-varime-z-naklicenych-lustenin/

-          Reference list needs to be completed by doi codes. I have added some doi codes in the specific comments but the authors should do it complete on the whole reference list, see e.g. reference numbers 137, 138, 139, 140, 141, 142, 144, 145 etc.

-          One other comment for reference list: year numbers should be written by bold but some of them are not bold.

-          Section 2 Food Proteins is placed on lines 221 and 385. There is no relation with main topic of the manuscript “Electrical Fields”. I recommend shortening this part and placing it into the paper supplement.

Specific comments

-          Line 154 contains reference Samaranayake et .al. (2006). There are two references in the reference list dated 2016 and 2018. There is need to correct reference in the text.

-          Line 355 contains Pernell et al (2022). Checking the reference on web true date is probably 2002 as it is given in the reference list.

-          Line 803 there is “Authors suggest that PEF”. There should be probably Gómez et al. (2019).

-          Line 814 There is missing [112]” after reference Dong et al. , 2021.

-          Line 831 Add after Caruggi et al. also year (2019) and [119]. Placing the citation number at the end of the paragraph is not appropriate.

-          Line 838 needs also complete reference Rodrigues et al. (2020) [121]. Placing the citation number at the end of the paragraph is not appropriate.

-          Lines 917-918 cite [142]. True reference in the text should be “Cao and Mezzenga (2019)”. There are not coworkers in the reference 142.

-          Line 934 input [126] after reference SubaÅ›i et al (2021). Otherwise this text reference is not related to full reference list.

-          Line 930 needs to shift [125] just behind the Pereira et al (2016). Placing the citation number at the end of the paragraph is not appropriate.

-          Line 1088 contains reference Cinzia et al. Such reference cannot be finding in the manuscript or in the references list. Reference [162] given on line 1091 is not related to this author.

-          Line 1091 should be completed by year (2016)

-          Line 1115 needs to revise reference in the text “Chen et al (2022) [128]. Placing the citation number at the end of the paragraph is not appropriate.

-          Line 1182 doi code is missing: https://doi.org/10.1016/j.foodres.2019.108586

-          Line 1187 reference 5 needs doi code: https://doi.org/10.1007/978-3-319-26779-1_194-1

-          Line 1188 reference 6 needs doi code: https://doi.org/10.1201/b16605

-          Line 1453 Reference 133 should have doi code: https://doi.org/10.1016/j.foodres.2019.04.047

-          Line 1459 doi code is missing: https://doi.org/10.1080/10408398.2022.2130154

-          Line 1492 exchange PH by pH.           

Author Response

We acknowledge the comments of the Reviewer 2 on the manuscript foods-2827923, which helped us to improve its overall quality. 

Authors comments can be found in the attachment. 

Reviewer 3 Report

Comments and Suggestions for Authors

The manuscript “Electrical Fields in the Processing of Protein-Based Foods” offers an in-depth and accurate overview of the structural and biological effects of electric field processing on protein fractions from various sources.

The manuscript is well written and structured.

Minor changes are suggested.

Lines 51-53: Which allergenic elements? Insert at least one reference.

Lines 70-73: Insert at least one reference.

Figure 3: Change clorela with chlorella

Author Response

We acknowledge the comments of the Reviewer 3 on the manuscript foods-2827923, which helped us to improve its overall quality. 

Authors comments can be found in the attachment. 

Reviewer 4 Report

Comments and Suggestions for Authors

 Review for Electrical Fields in the Processing of Protein-Based Foods  

By Ricardo N. Pereira, Rui Rodrigues, Zita Avelar, Ana Catarina Leite, Rita Leal and Ricardo S. Pereira, e António Vicente*

Delete e in  e António Vicente*

-------------

Electric-field based technologies offer interesting perspectives which include controlled heat dissipation (via the ohmic heating effect) and the influence of electrical variables (e.g. electroporation).

I agree that these factors collectively provide an opportunity to modify the functional and technological properties of numerous food proteins including the ones from emergent plant and microbial-based sources.

What is the penetration % of Electric-field based technologies in the food industry?

Electric-field based technologies present in one company among ten, five, two, one hundred?

-----------------------------

 The exploration of 40 alternative protein sources, such as plant-based proteins (derived from legumes, grains, 41 and vegetables), microorganisms (yeast, bacteria) algae, and insects, has gained traction. 42 These sources not only offer sustainable options but also diversify dietary choices and 43 contribute to reducing the environmental footprint of food production.

Very short presentation of the alternative protein sources

-------------------------

The need of dietary transition to other proteins sources like plant-based, microbial or 1131 insects) presents numerous technological challenges regarding their functional properties 1132 such as solubility, gelation, and emulsion among others

Fully true, your review is welcome

Author Response

We acknowledge the comments of the Reviewer 4 on the manuscript foods-2827923, which helped us to improve its overall quality. 

Authors comments can be found in the attachment. 
